# Anion Influence on Supramolecular Interactions in *Exo*-Coordinated Silver(I) Complexes with $N_2O_2$ Schiff Base Macrocycle

**Franc Perdih** [1] , **Milenko Korica** [2], **Lorena Šebalj** [2] **and Tomislav Balić** [2,*]

[1] Faculty of Chemistry and Chemical Technology, University of Ljubljana, Večna pot 113, SI-1000 Ljubljana, Slovenia
[2] Department of Chemistry, Josip Juraj Strossmayer University of Osijek, Cara Hadrijana 8/A, 31000 Osijek, Croatia
[*] Correspondence: tombalic@kemija.unios.hr; Tel.: +385-31-399-963

**Abstract:** Silver(I) complexes with aza-oxa macrocyclic Schiff bases L (L = 1,5-diaza-2,4:7,8:16,17-tribenzo-9,15-dioxa-cyclooctadeca-1,5-dien) were prepared by the reaction of the corresponding macrocycle with four different silver salts (AgX; X = $ClO_4$, $PF_6$, $SbF_6$ and $BF_4$). In all four compounds, silver ions are *exo*-coordinated by two neighboring ligand molecules in linear and T-shaped geometries. Such a coordination mode results in the formation of infinite 1D polymeric chains. Compounds $AgLClO_4$ and $AgLBF_4$ are isostructural, and polymeric chains display 1D zigzag topology. In $AgLPF_6$ there are three symmetrically unique Ag ions in the asymmetric unit of the compound. Two silver ions are linearly coordinated with two neighboring ligand molecules and are part of a discrete polymer chain. The third silver ion is coordinated with two ligand molecules and a methanol molecule in a T-shaped geometry. Such coordination geometry results in the formation of two discrete infinite polymer chains in the crystal structure. In the $AgLSbF_6$ compound, the chain topology is a linear zigzag chain, but in this compound, there is a difference in the orientation of the Ag-N bond. The Ag-N-Ag bonds are in the *trans* position relative to the plane calculated through the ligand molecule, while the Ag-N bonds are in the *cis* position in all other compounds. Due to the presence of a bulky $SbF_6$ anion, the ligand molecule is planar compared to other compounds. Considering intermolecular interactions, there is a huge variety of different interactions, mostly depending on the type of anion. A general supramolecular motif in all compounds is best described as 2D sheets of ligand–metal polymers with anions and solvent molecules sandwiched between them. In addition, the obtained compounds were characterized by IR spectroscopy and thermal analysis. The TG analysis indicates a rather surprising and considerable thermal stability of the prepared compounds, with some compounds thermally stable over 300 °C.

**Keywords:** *exo*-coordination; anions influence; supramolecular interactions; $N_2O_2$-donor; Schiff base macrocycles; silver(I) coordination polymers



## 1. Introduction

Macrocycles and macrocyclic compounds have been important parts of the investigation of hybrid organic–inorganic materials for almost six decades. Macrocycles, as organic ligands, can be used for the separation of metal cations, anions, and neutral molecules [1–5]. In order for macrocyclic ligands to bind successfully to certain chemical species, it is important to consider several different factors, such as the size of the macrocyclic ring, the properties of the donor atoms, the substituents in the ring, the properties of the metal ions (ionic radius, hardness or softness), the polarity of the system, and the orientation of donor atoms (*exo* or *endo*) [6–9]. The fundamental characteristic of macrocyclic complexes is increased thermodynamic stability compared to analogous acyclic complexes. This phenomenon is called the macrocyclic effect, which is the major reason for various applications

of macrocycles (e.g., sensors and selective extraction). Macrocyclic ligands have proven convenient to use since their cyclic nature limits ring flexibility as well as provides a central site that can be designed to support both the steric and electronic requirements of the metal ion. As a consequence of the above, the metal cation is most often bound in the macrocyclic cavity (*endo* coordination). Compounds in which *exo*-coordination is present (i.e., binding of a metal ion outside the cavity) are much rarer, but this type of coordination is certainly an interesting way to synthesize macrocyclic coordination polymers (CPs) [6,10–12]. Considering aza-oxa Schiff base macrocycles, previous investigations indicate that a certain level of ligand preorganization for the preparation of *exo*-coordinated species can be achieved by the presence of a rigid C=N bond near the aromatic system [6]. *Exo*-coordinated macrocyclic species are rarely reported, although these compounds are very attractive due to their structural diversity and as potential 3D porous materials [13–15]. In the context of a supramolecular assembly, anions can serve as an important tool for the design of CPs. In general, anions can be classified as coordinating species ($NO_3^-$, carboxylate), moderately coordinating ($CF_3SO_3^-$), and non-coordinating or weakly coordinated (i.e., $PF_6^-$, $ClO_4^-$, $BF_4^-$). All of these anions are commonly found in crystal structures of CPs with different structural roles: secondary building unit (SBU—usually in 3D polymers), building unit, templating, and charge balance (spectator) [16]. Anions can also be an important factor in the formation and overall topology of *exo*-coordinated macrocyclic species. It was found for silver(I) oxa-thia macrocyclic coordination compounds that the choice of the anion can promote the formation of an *exo*-coordinated (presence of $ClO_4^-$) and *endo*-coordinated complex (presence of $PF_6^-$) [17]. In addition, anions can be important when designing CPs for specific applications, such as energetic, magnetic, optical, and other properties [18–21]. Schiff bases are excellent ligands for the complexation of metal cations due to the π-acceptor character of the imine bond that leads toward the formation of highly stable metal complexes. Due to this character, Schiff bases are also prone to the formation of stable metal complexes with metal cations in lower oxidation states and filled *d* orbitals (i.e., Ag, Cu, Ni, Hg cations) [22,23]. Several reasons make Schiff bases excellent macrocyclic ligands: convenient preparation in high yields from aldehyde, ketone and amine precursors, and different degrees of self-condensation ([1 + 1], [2 + 2], [3 + 2], [4 + 4]) that can result in the formation of porous compounds, easy and efficient reduction in secondary amines, etc. [24]. The chemistry of coordination polymers with silver ion and *N*-donor ligands is very well-known and has been widely researched during the last couple of decades [25]. Perhaps the most important reason for such a vast number of investigations is the very flexible coordination sphere of silver cation that allows the formation of CPs with different dimensionalities (1D, 2D, 3D) and coordination numbers [26,27]. Such structural flexibility is extremely important in the rational design of silver-based CPs as active pharmaceutical ingredients (APIs) [28,29].

Herein we report the synthesis of four novel silver(I) complex compounds with an N2O2-donor macrocyclic ligand (L). The compounds were identified by chemical analysis and characterized by IR spectroscopy and thermal analysis. The molecular and crystal structures were determined by single-crystal X-ray diffraction. Herein, reported structures were compared to previously reported nitrate analog (CSD code BUPREL) [30] and the impact of anions on the supramolecular features of compounds was discussed.

## 2. Materials and Methods

### 2.1. General Methods

Silver salts were purchased from Sigma-Aldrich (98% purity) and used as purchased. Organic solvents used for syntheses were obtained from different commercial sources. All solvents were p.a. quality and used as purchased without additional purification. IR spectra measurements were done on a Shimadzu FTIR 8400S spectrophotometer using the DRS 8000 attachment, in the 4000–400 cm$^{-1}$ region. A total of 3 mg of samples was mixed with 100 mg of KBr (IR grade) and measured in the sample cup. The data were collected using the diffuse reflectance technique. Thermogravimetric analyses were performed using a

simultaneous TGA-DSC analyzer (Mettle-Toledo TGA/DSC 1). Approximately 5 to 10 mg of samples were placed in 100 μL aluminum pans, heated in an oxygen atmosphere (195 mL min$^{-1}$) up to 550 °C at a rate of 10 °C min$^{-1}$. The data collection and analysis were performed using the program package STARe Software 10.0 [31]. PerkinElmer 2400 Series II CHNS/O system was used for C, H, N analyses.

*2.2. X-Ray Crystallography*

Crystals of all compounds decomposed when filtered from the mother liquor and were therefore transferred into mineral oil (Paratone N) and scooped with a plastic loop. Diffraction data were collected at 150 K on an Agilent Technologies SuperNova Dual diffractometer using (Mo-Kα radiation, λ = 0.71073 Å). The data reduction was performed using the CrysAlis software package [32]. The structures were solved with the ShelXT program [33] and refined by a full-matrix least-squares procedure based on $F^2$ with ShelXL [34] using Olex2 program suite [35]. All non-hydrogen atoms were refined anisotropically. Hydrogen atoms in the structures were placed in calculated positions and refined using the riding model. In AgLClO$_4$, perchlorate atoms Cl1, O3, and O6 were disordered over two positions with the refined ratio 0.66(6):0.34(6). In AgLPF$_6$ atoms F15–F18, part of one PF$_6^-$ anion was disordered over two positions with a refined ratio 0.67(3):0.33(3), and methanol O8 oxygen atom was disordered over two positions with a fixed occupancy of 0.55:0.45. Atoms O8B and F17B were refined restraining Uij components. A high residual electron density between adjacent polymeric chains was found in AgLSbF$_6$ during refinement. Unfortunately, all attempts to assign electron density to specific chemical species failed, presumably due to heavily disordered solvent molecules. The structure was further analyzed using the BYPASS subroutine implemented in Olex2 and the final refinement data (Table 1) is given without residual electron density. The BYPASS analysis indicates the presence of 49 electrons per formula unit that can be assigned to approximately 1 dichloromethane molecule. PLATON [36] was used for geometrical calculations and MERCURY [37] for the structure drawings. The crystallographic data are summarized in Table 1. The selected bond lengths and angles are presented in ESI (Tables S1–S4).

*2.3. Synthesis*

Macrocyclic ligand L was prepared by the previously published method [30,38].

1.  Preparation of the silver(I) coordination polymers

All CPs were prepared using the following procedure: 0.05 mmol of ligand was dissolved in 4 mL of dichloromethane. A total of 0.05 mmol of appropriate silver salt was dissolved in 4 mL of methanol. In the U-shaped tube, 2 mL of chloroform was added and the prepared ligand and metal salt solutions were gradually added. After a few days, transparent single crystals suitable for diffraction experiments formed on the edges of the tube.

2.  Preparation of AgLPF$_6$

Used: AgPF$_6$ (12.6 mg, 0.05 mmol in 20 mL), L (19.2 mg; 0.05 mmol). Yield: 27 mg, 90% (based on the silver salt). Anal. Calc. for: C$_{25}$H$_{24}$AgN$_2$O$_2$PF$_6$ C, 47.12; H, 3.8; N, 4.4. Found: C, 47.32, H, 3.61; N, 4.68; IR ν$_{max}$ (cm$^{-1}$): 3361(m), 3570(m), 1602(s), 831(s), 557(s).

3.  Preparation of AgLSbF$_6$

Used: AgSbF$_6$ (17.2 mg, 0.05 mmol in 20 mL), L (19.2 mg; 0.05 mmol). Yield: 25 mg, 69% (based on the silver salt). Anal. Calc. for: C$_{25}$H$_{24}$AgN$_2$O$_2$SbF$_6$ C, 41.24; H, 3.32; N, 3.85. Found: C, 41.05, H, 3.28; N, 3.68; IR ν$_{max}$ (cm$^{-1}$): 1614(s), 630(m).

4.  Preparation of AgLBF$_4$

Used: AgBF$_4$ (9.7 mg, 0.05 mmol in 20 mL), L (19.2 mg; 0.05 mmol). Yield: 29 mg, 82% (based on the silver salt). Anal. Calc. for: C$_{25}$H$_{24}$AgN$_2$O$_2$BF$_4$ C, 51.85; H, 4.18; N, 4.84. Found: C, 52.01, H, 4.28; N, 4.58; IR ν$_{max}$ (cm$^{-1}$): 1602(m), 1055(s).

## 5. Preparation of AgLClO$_4$

Used: AgClO$_4$ xH$_2$O(10.3 mg, 0.05 mmol in 20 mL), L (19.2 mg; 0.05 mmol). Yield: 27 mg, 77% (based on the silver salt). Elemental analysis was not performed due to the potentially explosive nature of the compound; IR $\nu_{max}$ (cm$^{-1}$): 1616(m), 1109(s), 1082(s), 621(s), 420(w).

**Table 1.** Crystallographic data and structure refinement details for all compounds.

| Compound | AgLClO$_4$ | AgLPF$_6$ | AgLBF$_4$ | AgLSbF$_6$ |
|---|---|---|---|---|
| CCD number | 2223301 | 2223302 | 2223300 | 2223303 |
| Empirical formula | C$_{26}$H$_{25}$AgCl$_4$N$_2$O$_6$ | C$_{77}$H$_{80}$Ag$_3$F$_{18}$N$_6$O$_8$P$_3$ | C$_{26}$H$_{24}$AgBCl$_3$F$_4$N$_2$O$_2$ | C$_{25}$H$_{23}$AgF$_6$N$_2$O$_2$Sb |
| Formula weight | 711.15 | 1975.99 | 697.50 | 727.092 |
| Temperature/K | 150.00(10) | 150.00(10) | 150.00(10) | 150.00(10) |
| Crystal system | orthorhombic | orthorhombic | orthorhombic | monoclinic |
| Space group | P2$_1$2$_1$2$_1$ | P2$_1$2$_1$2$_1$ | P2$_1$2$_1$2$_1$ | P2$_1$/n |
| a/Å | 8.2849(4) | 14.6441(4) | 8.2608(7) | 10.4581(3) |
| b/Å | 15.7717(6) | 21.4869(5) | 15.7088(6) | 23.7924(9) |
| c/Å | 21.1305(8) | 24.9257(10) | 21.0174(10) | 11.9083(6) |
| α/° | 90 | 90 | 90 | 90 |
| β/° | 90 | 90 | 90 | 95.943(4) |
| γ/° | 90 | 90 | 90 | 90 |
| Volume/Å$^3$ | 2761.1(2) | 7843.0(4) | 2727.4(3) | 2947.1(2) |
| Z | 4 | 4 | 4 | 4 |
| ρ$_{calc}$g/cm$^3$ | 1.711 | 1.673 | 1.699 | 1.639 |
| μ/mm$^{-1}$ | 1.162 | 0.905 | 1.089 | 1.641 |
| F(000) | 1432.0 | 3984.0 | 1396.0 | 1415.1 |
| Crystal size/mm$^3$ | 0.2 × 0.15 × 0.1 | 0.25 × 0.2 × 0.1 | 0.2 × 0.2 × 0.2 | 0.35 × 0.25 × 0.2 |
| Radiation | MoK$\alpha$ ($\lambda$ = 0.71073) | MoK$\alpha$ ($\lambda$ = 0.71073) | MoK$\alpha$ ($\lambda$ = 0.71073) | Mo K$\alpha$ ($\lambda$ = 0.71073) |
| 2Θ range for data collection/° | 4.64 to 54.97 | 4.702 to 54.966 | 4.664 to 54.968 | 4.86 to 54.96 |
| Index ranges | $-9 \leq h \leq 10$, $-20 \leq k \leq 19$, $-25 \leq l \leq 27$ | $-13 \leq h \leq 19$, $-27 \leq k \leq 27$, $-19 \leq l \leq 32$ | $-7 \leq h \leq 10$, $-20 \leq k \leq 20$, $-26 \leq l \leq 27$ | $-14 \leq h \leq 13$, $-32 \leq k \leq 33$, $-16 \leq l \leq 16$ |
| Reflections collected | 13345 | 36345 | 9867 | 27761 |
| Independent reflections | 6328 [R$_{int}$ = 0.0322, R$_{sigma}$ = 0.0497] | 17962 [R$_{int}$ = 0.0267, R$_{sigma}$ = 0.0455] | 5858 [R$_{int}$ = 0.0251, R$_{sigma}$ = 0.0512] | 6762 [R$_{int}$ = 0.0300, R$_{sigma}$ = 0.0318] |
| Data/restraints/parameters | 6328/0/380 | 17962/12/1088 | 5858/0/352 | 6762/48/371 |
| Goodness-of-fit on F$^2$ | 1.069 | 1.030 | 1.062 | 1.077 |
| Final R indexes [I >= 2σ (I)] | R$_1$ = 0.0541, wR$_2$ = 0.1421 | R$_1$ = 0.0373, wR$_2$ = 0.0739 | R$_1$ = 0.0566, wR$_2$ = 0.1527 | R$_1$ = 0.0449, wR$_2$ = 0.1067 |
| Final R indexes [all data] | R$_1$ = 0.0667, wR$_2$ = 0.1551 | R$_1$ = 0.0515, wR$_2$ = 0.0815 | R$_1$ = 0.0697, wR$_2$ = 0.1660 | R$_1$ = 0.0569, wR$_2$ = 0.1163 |
| Largest diff. peak/hole/e Å$^{-3}$ | 1.26/−1.41 | 1.04/−0.68 | 1.40/−1.72 | 1.01/−1.05 |
| Flack parameter | 0.01(2) | 0.297(19) | −0.01(2) | |

$R = \sum ||F_o| - |F_c|| / \sum |F_o|$. $wR_2 = \{\sum[w(F_o{}^2 - F_c{}^2)^2] / \sum[w(F_o{}^2)^2]\}^{1/2}$.

### 3. Results and Discussion

#### 3.1. Synthesis, Structure of the Ligand (L), and IR Spectroscopy

Ligand L was prepared by a cyclocondensation reaction of dialdehyde and diamine in high yields. Ligand L is an 18-membered $N_2O_2$ donor macrocyclic Schiff base with N-donor atoms *exo*-oriented and O-donor atoms *endo* oriented. The molecule is non-planar with large deviations between benzene rings and puckering amplitude of the inner macrocyclic ring of 1.843(3) Å (Table S5). Considering donor atoms and conformation, the ligand is suitable for the formation of *exo*-coordinated CPs. Complex coordination polymers were obtained by diffusion crystallization, using a methanol solution of different silver(I) salts and a dichloromethane solution of ligand in the U-shaped tube. Chloroform was used as a barrier between the two solutions. Crystals suitable for diffraction measurements appeared on the edges of a U-shaped tube (on the side with the ligand solution) in approximately one to two days. After filtration and in contact with air, crystals of all compounds become opaque and lose their crystallinity, most probably due to the evaporation of included solvents. In all compounds, polymeric structures are formed with anions that are located between discrete polymer chains and act as counterions (Scheme 1).

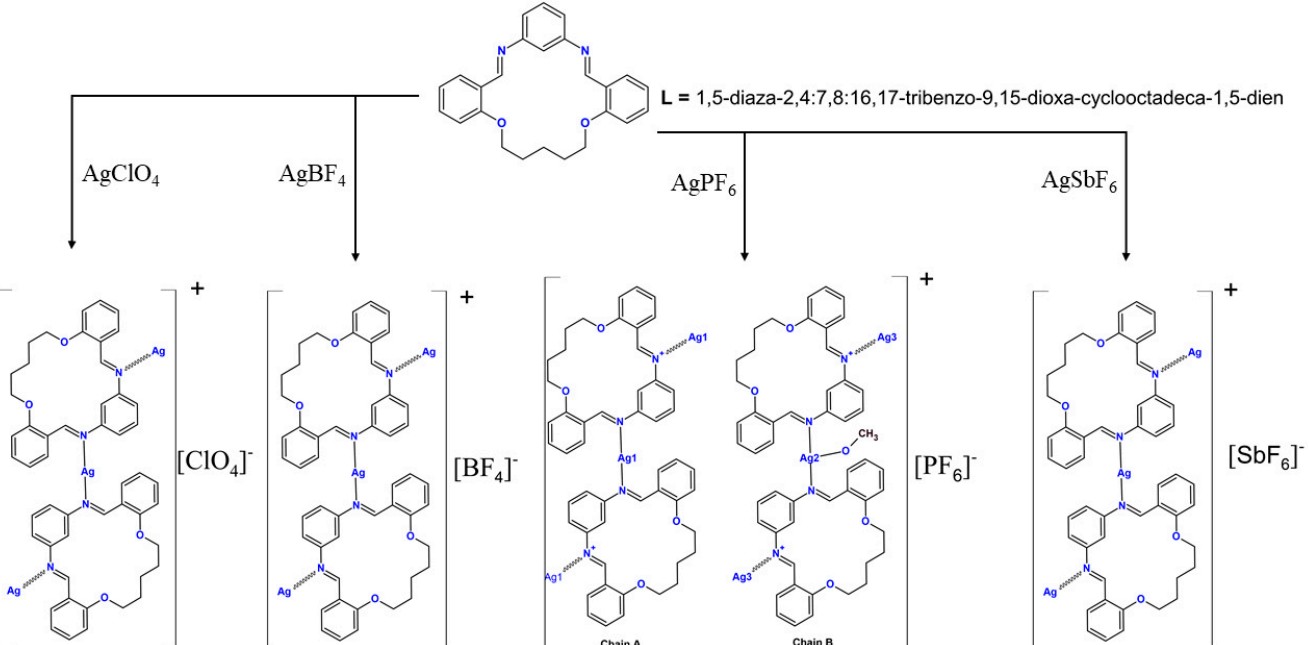

**Scheme 1.** Synthesis pathway for the preparation of silver complex compounds with $ClO_4$, $BF_4$, $SbF_6$, and $PF_6$ anions.

Spectra of synthesized coordination polymers are similar (Figures S1–S4, see in Supplementary Materials). A common characteristic in all spectra of CPs is a slight redshift of imine stretching vibrations in comparison to the spectrum of L [30]. This shift indicates the coordination of ligand to silver ion. In the spectrum of AgLClO$_4$, vibrations close to 1100 cm$^{-1}$ and 620 cm$^{-1}$ can be assigned to the perchlorate anion [6]. The presence of the BF$_4$ anion can be seen in the spectrum of AgLBF$_4$ as a broad maximum at approximately 1050 cm$^{-1}$, which is typical for this anion [39]. In spectra of AgLSbF$_6$ and AgLPF$_6$, vibrations at 630 cm$^{-1}$ and 830 cm$^{-1}$ are assigned to SbF$_6$ and PF$_6$ anions [40,41], respectively. Other vibrations typical for the ligand molecule are also observed in the spectra of CPs.

#### 3.2. Crystal Structures of Ag(I) Coordination Polymers

Compounds AgLClO$_4$, AgLPF$_6$, and AgLBF$_4$ crystallize in the orthorhombic crystal system, space group $P2_12_12_1$ and AgLSbF$_6$ in the monoclinic crystal system, and the $P2_1/n$ space group. In AgLClO$_4$, silver ions are coordinated by two ligand molecules in linear

geometry (N1–Ag1–N2#1 angle of 175.32(19)°) (Figure 1a). Such coordination produces an infinite polymeric structure along the crystallographic axis b, and the topology of the polymeric chain is a 1D zigzag chain. Perchlorate ions and chloroform molecules are sandwiched between two adjacent polymeric chains. Both perchlorate ions and chloroform molecules are connected mutually and with polymeric chains by several weak C–H···O interactions (Figure 1b and Table 2) along the a axis.

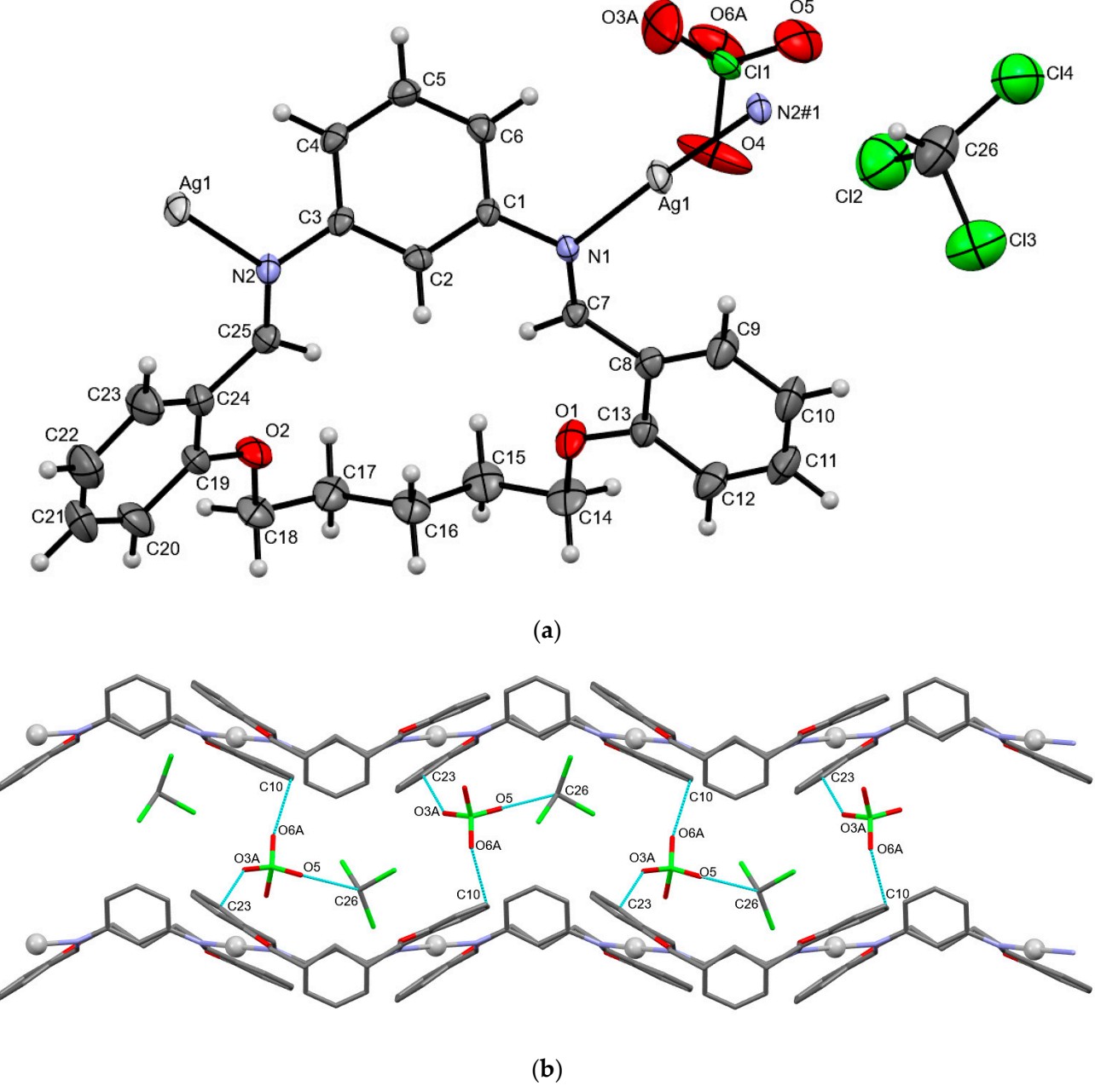

(a)

(b)

**Figure 1.** (**a**) ORTEP plot of AgLClO$_4$ with displacement ellipsoids of non-hydrogen atoms drawn at the 50% probability level. Symmetry operation: #1 $1 - x$, $-1/2 + y$, $1/2 - z$. (**b**) Representation of C–H···O intermolecular interactions (light blue lines) in AgLClO$_4$ (hydrogen atoms are omitted for clarity and silver ions are represented as spheres). The view is down the *a* axis.

The complex AgLBF$_4$ is isostructural to previously described AgLClO$_4$ (Figure 2a). There are some minor differences in bond lengths and angles and conformation of ligand molecules. Surprisingly, the supramolecular motif(s) in this compound is almost identical to AgLClO$_4$: 1D *zigzag* polymeric chains along the *b* axis with chloroform molecules and

BF$_4$ anions trapped between polymeric chains (along the *a* axis). Chloroform molecules and BF$_4$ anions are connected by weak C–H···F interactions and to polymeric chains by weak C–H···F and Cl···π interactions (Cg2→C8–C13 benzene ring) (Figure 2b and Table 2).

**Table 2.** The details on hydrogen bond geometry for AgLClO$_4$, AgLBF$_4$, AgLPF$_6$, and AgLSbF$_6$. π···π interactions in AgLSbF$_6$, Y–X···Cg in AgLBF$_4$ and AgLPF$_6$, and X–H···Cg in AgLPF$_6$.

| **AgLClO$_4$** | | | | | |
|---|---|---|---|---|---|
| | *d*(D–H) | *d*(H···A) | *d*(D···A) | ∠(D–H···A) | symmetry operator |
| C23–H23···O3A | 0.95 | 2.193 | 2.935(7) | 134(9) | −x + 1, +y + 1/2, −z + 1/2 |
| C26–H26···O5 | 1.00 | 2.236 | 3.107(1) | 144(8) | x, y, z |
| C10–H10···O6A | 0.95 | 2.881 | 3.072(1) | 92(1) | x, y, z |
| **AgLBF$_4$** | | | | | |
| | *d*(D–H) | *d*(H···A) | *d*(D···A) | ∠(D–H···A) | symmetry operator |
| C26–H26···F4 | 1.00 | 2.208 | 3.087(7) | 145(8) | x, y, z |
| Y–X···Cg | X···Cg (Å) | Y···Cg | γ | Y–X···Cg | symmetry operator |
| C26–Cl2···Cg2(C8→C13) | 3.573(1) | 5.213(1) | 16.36 | 159(3) | x, y, z |
| **AgLPF$_6$** | | | | | |
| | *d*(D–H) | *d*(H···A) | *d*(D···A) | ∠(D–H···A) | symmetry operator |
| O7–H7A···F18A | 0.84 | 2.08 | 2.906(4) | 169(1) | −1/2 + x, 1/2 − y, 1 − z |
| O8A–H8A···F7 | 0.84 | 2.04 | 2.849(7) | 163(1) | x, y, z |
| C6–H6···F7 | 0.95 | 2.42 | 3.144(7) | 133(1) | 1/2 + x, 1/2 − y, −z |
| C20–H20···F16A | 0.95 | 2.39 | 3.201(9) | 143(1) | 1/2 − x, −y, −1/2 + z |
| C29–H29···F6 | 0.95 | 2.30 | 3.054(7) | 136(1) | −1 + x, y, z |
| C37–H37···F3 | 0.95 | 2.41 | 3.341(3) | 167(1) | 1 − x, −1/2 + y, 1/2 − z |
| C70–H70···F10 | 0.95 | 2.48 | 3.403(9) | 165(1) | 1 − x, 1/2 + y, 1/2 − z |
| C42–H42B···F11 | 0.99 | 2.44 | 3.384(8) | 160(1) | −x, −1/2 + y, 1/2 − z |
| Y–X···Cg | X···Cg (Å) | Y···Cg | γ | Y–X···Cg | symmetry operator |
| P1–F5···Cg7(C1→C6) | 3.4064 | 4.3990 | 27.29 | 119 | x, y, z |
| P2–F12···Cg4(C51→C56) | 3.2838 | 4.3559 | 20.63 | 123 | x, y, z |
| P3–F15A···Cg1(C26→C31) | 3.2308 | 4.4960 | 20.84 | 133 | 1/2 + x, 1/2 − y, 1 − z |
| X–H···Cg | X···Cg (Å) | H···Cg | γ | X–H···Cg | symmetry operator |
| C68–H68A···Cg9(C19→C24) | 3.5634 | 2.85 | 12.84 | 130 | 1 − x, 1/2 + y, 1/2 − z |
| **AgLSbF$_6$** | | | | | |
| | *d*(D–H) | *d*(H···A) | *d*(D···A) | ∠(D–H···A) | symmetry operator |
| C6–H6···F3A | 0.95 | 2.57 | 3.463(1) | 156(3) | x + 1/2, −y + 1/2 + 1, +z − 1/2 |
| C9–H9···F6A | 0.95 | 2.45 | 3.397(1) | 171(8) | x + 1/2, −y + 1/2 + 1, +z − 1/2 |
| C14–H14B···F4A | 0.99 | 2.39 | 3.354(1) | 163(4) | −x + 2, −y + 1, −z + 2 |
| C20–H20···F4A | 0.95 | 2.54 | 3.486(1) | 170(8) | −x + 1, −y + 1, −z + 2 |
| | Cg···Cg (Å) | α | β | Cg···plane | symmetry operator |
| Cg2 (C8→C13)··· Cg3 (C19→C24) | 3.629(4) | 6.296 | 21.41 | 3.445 | −1 + x, y, z |
| Cg3 (C19→C24) ···Cg2 (C8→C13) | 3.629(4) | 6.296 | 18.67 | 3.385 | 1 + x, y, z |

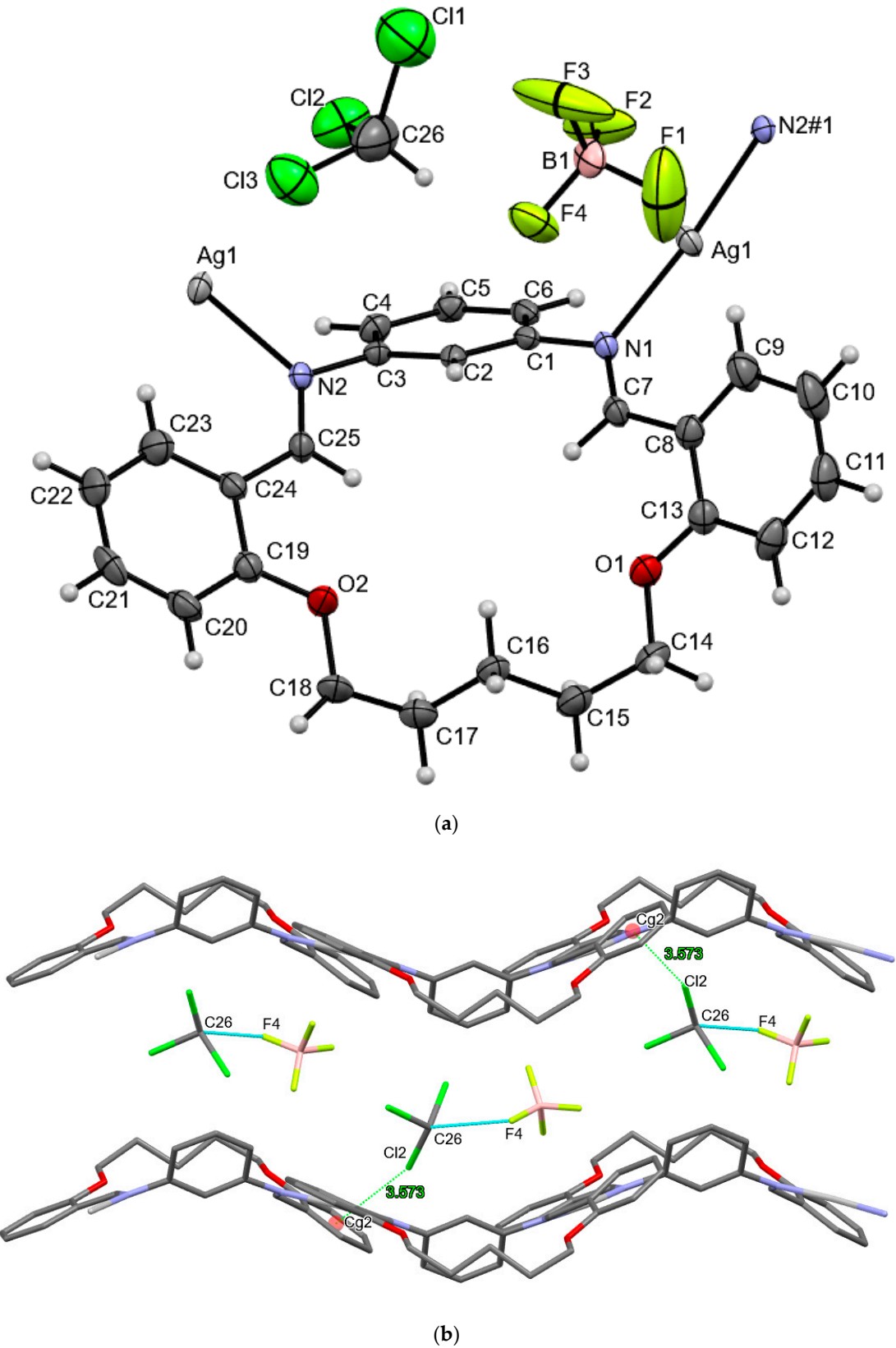

(a)

(b)

**Figure 2.** (**a**) ORTEP plot of AgLBF$_4$ with displacement ellipsoids of non-hydrogen atoms drawn at the 50% probability level. Symmetry operation: #1 $1 - x$, $1/2 + y$, $3/2 - z$. (**b**) Representation of C–H⋯F intermolecular interactions (light blue lines) and Cl⋯$\pi$ interactions (green dashed lines) in AgLBF$_4$. Hydrogen atoms are omitted for clarity.

In the asymmetric unit of the $AgLPF_6$ compound, there are three symmetrically unique Ag ions (Ag1, Ag2 and Ag3). The Ag2 and Ag3 ions are part of the discrete polymeric chain (Figure 3) and are not related by symmetry. The Ag2 ions are coordinated by two ligand molecules (symmetry unrelated) and additionally by a methanol molecule, in T-shaped geometry. The Ag3 ions are coordinated by two adjacent ligand molecules (symmetry-related) in a linear geometry (Figure 3). This mode of coordination produces an infinite polymeric chain along the a axis with $[Ag_2(MeOH)(L)_2]_n$ repeating units. The coordination of the Ag1 ion is almost identical to that previously described. A minor difference is observed in the N–Ag–N angle (N1–Ag1–N2#1 angle of 165.74(16)°) in comparison to $ClO_4$ and $BF_4$ analogs, and this might be explained by the presence of a bulkier $PF_6$ ion (see Table 3 and discussion regarding anion influence) near this bond. The topology of both polymeric chains can be described as a 1D zigzag chain. Both chains propagate along the crystallographic axis *a*.

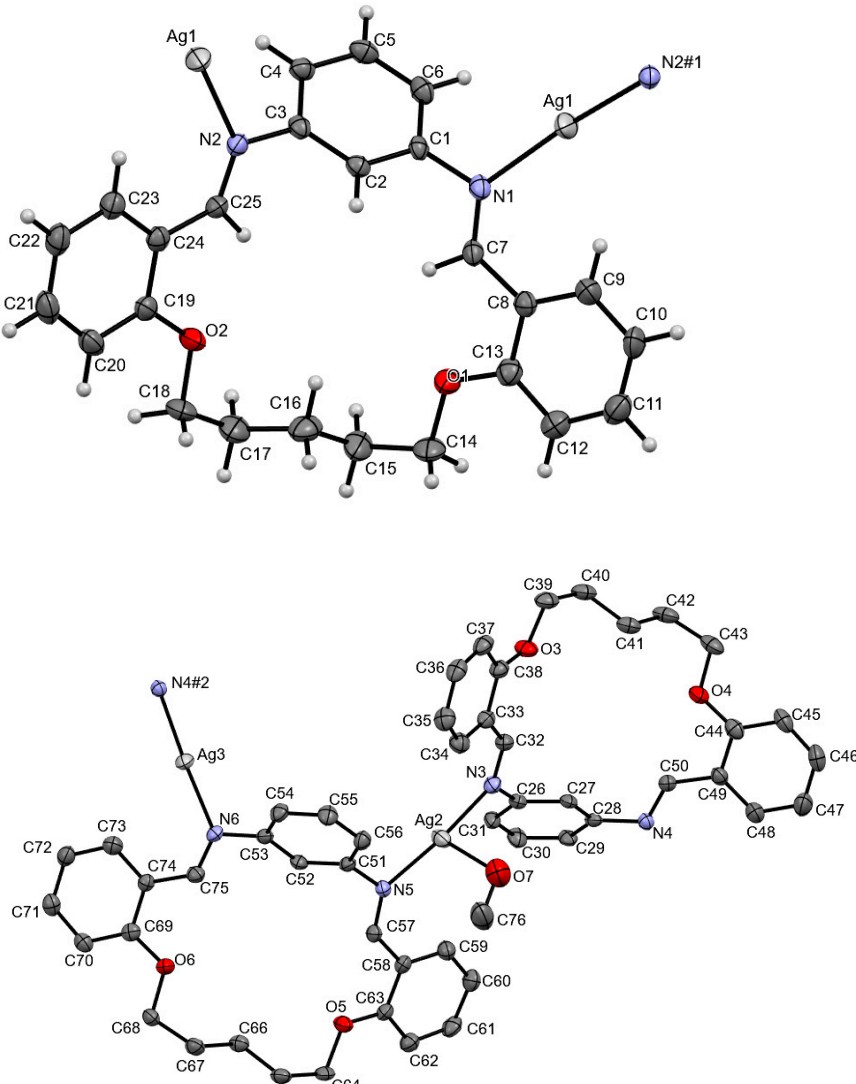

**Figure 3.** ORTEP plot of $AgLPF_6$ with displacement ellipsoids of non-hydrogen atoms drawn at the 50% probability level. Symmetry operation: #1 −1/2 + x, 3/2 − y,1 − z, #2 −1 + x, +y, +z. Hydrogen atoms, solvent methanol molecule, and $PF_6$ are omitted for the sake of clarity.

In the crystal, discrete polymeric chains are connected to $PF_6$ anions by O–H⋯F and P–F⋯$\pi$ interactions (Figure 4). Polymeric chain B a is connected to anions via coordinated MeOH molecules (O7–H7A⋯F18A) and P–F⋯$\pi$ interactions involving Cg1 and Cg4 ben-

zene rings (Cg1→C26–C31 and Cg4→C51–C56 benzene rings). Solvent MeOH molecules (uncoordinated) form weak O–H···F interactions with PF$_6$ anions (O8A–H8A···F7). PF$_6$ anions form P–F···π interactions with chain A benzene rings (Cg7→C1–C6). Adjacent polymeric chains are connected by a series of weak C–H···F interactions along crystallographic axis c. The final 3D arrangement of molecules is achieved by a series of weak C–H···F and C–H···π interactions along the *b* axis (Table 2). These interactions involve aliphatic chains, benzene rings, and PF$_6$ anions.

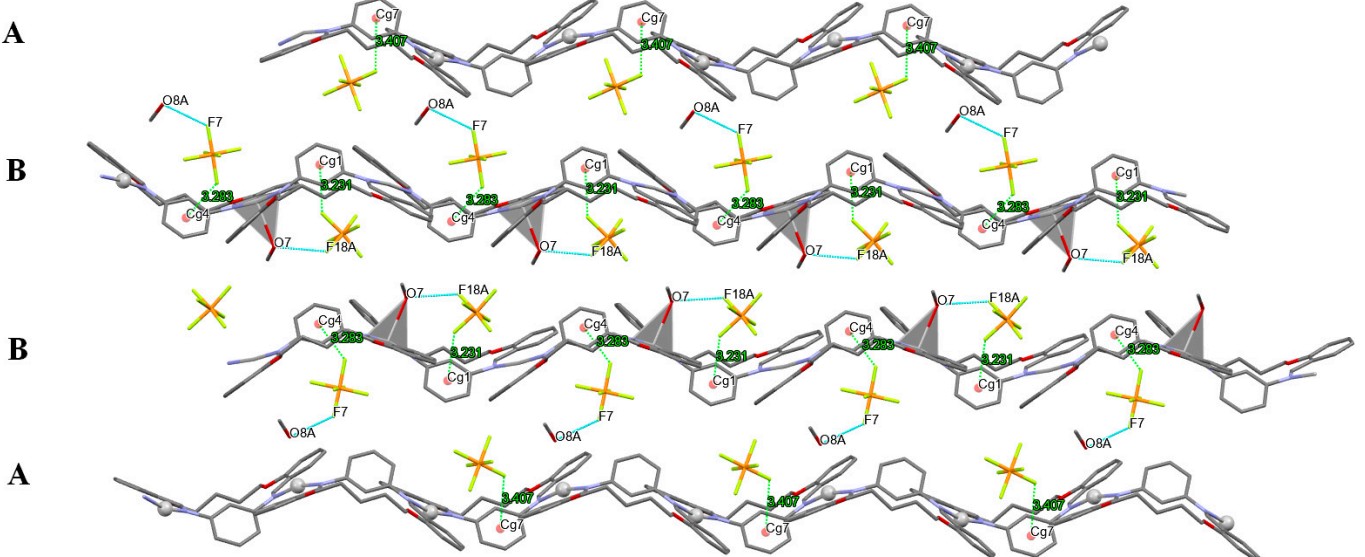

**Figure 4.** Representation of crystal packing in AgLPF$_6$ along the *c* axis. F···π interactions between PF$_6$ anion and benzene rings are represented by green dashed lines. O–H···F interactions between MeOH molecules (coordinated and uncoordinated) and discrete polymeric chains are represented by blue dashed lines. The view is down the *b* axis.

The molecular structure of AgLSbF$_6$ is shown in Figure 5. Considering coordination mode and chain topology, the compound is rather similar to that previously described (AgLClO$_4$ and AgLBF$_4$). As in previous structures, the chain topology can also be described as linear zigzag. However, there is a difference in the orientation of Ag–N bond propagation in this compound. If one considers the nitrogen atoms in the macrocyclic ring as the starting point of the polymer chain and calculates a plane through the ligand molecule, in the AgLSbF$_6$ it can be seen that Ag–N–Ag bonds are in the *trans* position (*trans* configured) to the calculated plane (Figure S6). In all other compounds, Ag–N–Ag bonds are in the *cis* position (*cis* configured). These differences are also observable in values of C–N–Ag–N torsion angles (Table S5). A major difference in the conformation of ligand molecules was also found in this compound. In comparison to other compounds, in AgLSbF$_6$ the ligand molecule is planar with a small deviation in angles between benzene rings (Table S5). The overlay of molecular structures can be found in ESI (Figure S5).

Two adjacent polymeric chains are connected by π···π interactions between benzene rings (Cg2 (C8→C13) ··· Cg3 (C19→C24)) approximately along [1–11] crystallographic directions. These aromatic interactions are observed only in AgLSbF$_6$ and are the consequence of macrocyclic ring flattening. SbF$_6$ anions are connected to polymeric chains via several weak C–H···F interactions (Figure 6 and Table 2). Such formed 2D supramolecular structures are connected by weak C–H···F interactions that involve carbon atoms of an aliphatic chain and SbF$_6$ anions approximately along the *c* axis.

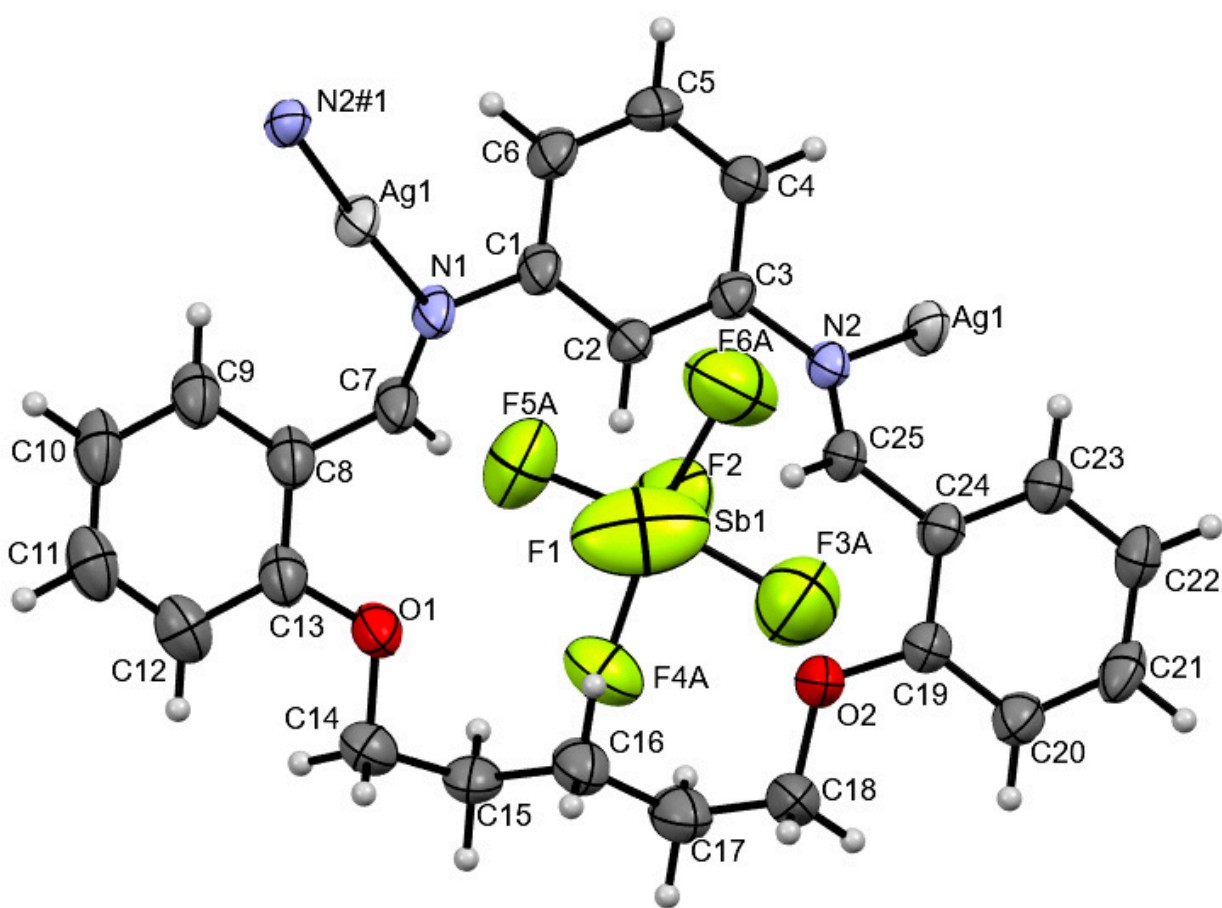

**Figure 5.** ORTEP plot of AgLSbF$_6$ with displacement ellipsoids of non-hydrogen atoms drawn at the 50% probability level. Symmetry operation: #1 $\frac{1}{2}$ + x, 3/2 − y, −1/2 + z. Disordered dichloromethane and chloroform molecules are omitted for clarity.

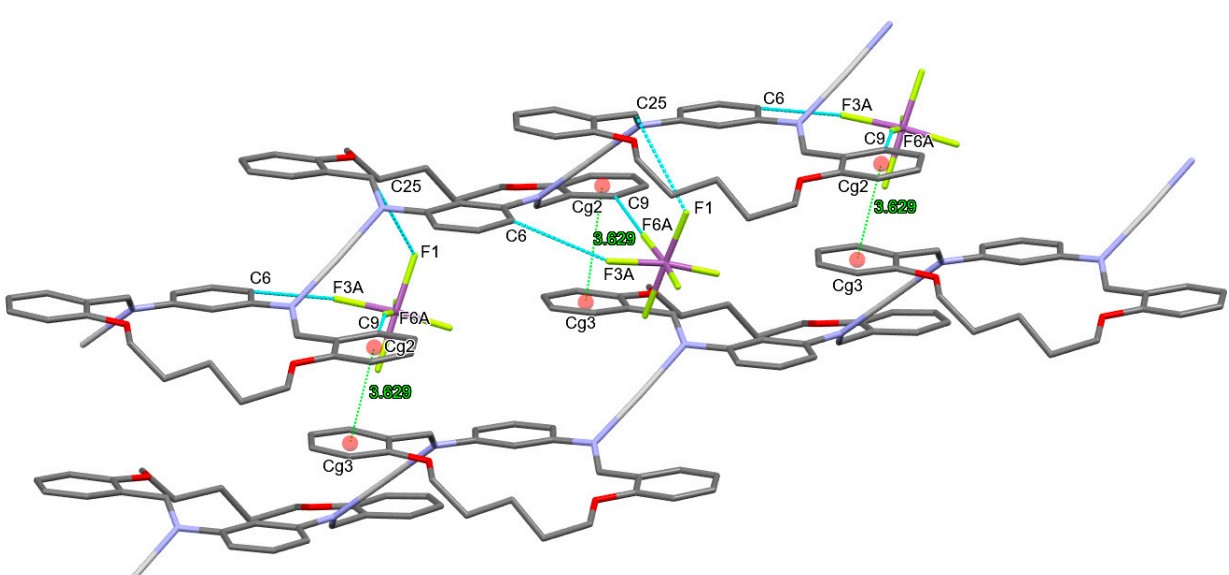

**Figure 6.** Representation of crystal packing in AgLSbF$_6$. O–H···F interactions between SbF$_6$ anions and ligand molecules are represented by blue dashed lines. $\pi$···$\pi$ interactions are represented by blue dashed lines. Hydrogen atoms are omitted for clarity.

### 3.3. Anion Influence on Supramolecular Assembly

In our previous publications [6,30], we have attempted to explain the influence of both anion and ligand structural aspects on the formation of coordination polymers with aza-oxa macrocycles. These investigations were limited to $NO_3$ and $ClO_4$ anions; therefore, a complete understanding of anion impact on supramolecular interactions was not achieved. Our previous observations can be briefly summarized: (i) ligand molecule (L) is essentially a non-planar molecule with O-donor atoms oriented in an *endo* manner and N-donor atoms in an *exo* manner, (ii) orientation of donor atoms does not change upon formation of coordination compounds, and (iii) some minor changes in ligand conformation (planarity and aliphatic chain conformation) are possible, but a reorientation of donor atoms is very unlikely. Considering ligand structural features, it was quite predictable that reactions with a silver(I) will form polymeric species, regardless of the presence of a different anion, which was confirmed in this study. To elucidate the influence of anions on the supramolecular structure, the nitrate-containing complex [30] with the same ligand (CSD code BUPREL) was included in the discussion. In this compound coordination, geometry is T-shaped, with nitrate anion monodentatelly coordinated to silver ion. Polymeric chain topology is also zigzag and adjacent chains are connected via series C–H···O interactions that involve uncoordinated oxygen atoms of nitrate anion (Figure 7). From the description of the supramolecular features of all of these compounds, a general supramolecular motif can be observed: 2D ribbon-like supramolecular network with anions and solvent molecules sandwiched between adjacent polymeric chains. Although this motif is present in all structures, some interesting differences can be ascribed to anion influence. $NO_3$, $BF_4$ and $ClO_4$ (smaller trigonal and tetrahedral anions—see Table 3) act as a bridge between adjacent polymeric chains. $NO_3$ anion in BUPREL is coordinated to the silver(I) and forms numerous weak C–H···O interactions with ligand molecules, thus emphasizing the importance of nitrate anion in supramolecular assembly. Considering the $BF_4$ anion, the most important interaction is with the solvent molecule, and in this case the solvent molecule (to be more precise, the Cl···$\pi$ interaction) contributes the most to the connection between adjacent chains. On the other hand, $ClO_4$ anion forms several ligand···anion interactions (C–H···O intermolecular interactions) between chains, and the solvent···ligand interactions were not observed; therefore, it can be argued that in this case the anion is responsible for the interchain connection. As a brief comment for these compounds, it can be stated that ligand molecule with its donor groups is more prone to the formation of X–H···O interactions than X–H···F. Recent investigation [42] on the $BF_4$ anion as a hydrogen bond donor or acceptor has shown that different types of bonds are possible; however, these bonds can be geometrically disrupted due to the tetrahedral shape of the anion. A similar conclusion can be drawn for $AgLBF_4$.

**Table 3.** Thermochemical radii and volume of anions used for the preparation of CPs.

| Anion | $NO_3$ | $BF_4$ | $ClO_4$ | $PF_6$ | $SbF_6$ |
|---|---|---|---|---|---|
| Thermochemical radii (nm) [43] | 200 | 205 | 225 | 242 | 252 |
| Molecular volume ($\text{Å}^3$) [44] | 34 | 38 | 47 | 54 | 71 |

The inclusion of larger octahedral anions ($PF_6$ and $SbF_6$) in crystal structures increases the number of intermolecular interactions. In both compounds, anions affect supramolecular assembly in a rather unique manner. In the $PF_6$ compound, there are two discrete polymeric chains (chains A and B in Figure 4) connected by a series of different interactions involving both solvent and anion molecules. What is especially interesting is that a different solvent molecule (methanol) is present in this structure. The methanol molecule is coordinated to the silver ion in chain B and additionally as a solvent between adjacent chains. Here, we were particularly interested in explaining why methanol is present in this structure. Two possibilities that would favor methanol over the other two solvents are the following: either the presence of methanol is due to the size of methanol (size

effect), or there is a "special" relationship between methanol and the $PF_6$ anion. To test our hypothesis, solvent methanol molecules were removed from the structure and BYPASS analysis was done. The results indicate that there is a void of 63 Å$^3$ in the structure when methanol molecules are omitted. A simple calculation of solvent molecular volumes using the ChemSketch program [45] showed that volumes are 42.5 Å$^3$ (methanol), 67.8 Å$^3$ (dichloromethane), and 79.5 Å$^3$ (chloroform). This simple calculation indicates that the size effect is most probably the reason for the presence of methanol in the structure, and in this case emphasizes the importance of anion size in the supramolecular assembly. Another interesting conclusion can be stated here: if these compounds are discriminatory towards solvents by size and dependent on anion size, then a plausible 3-step mechanism can be proposed:

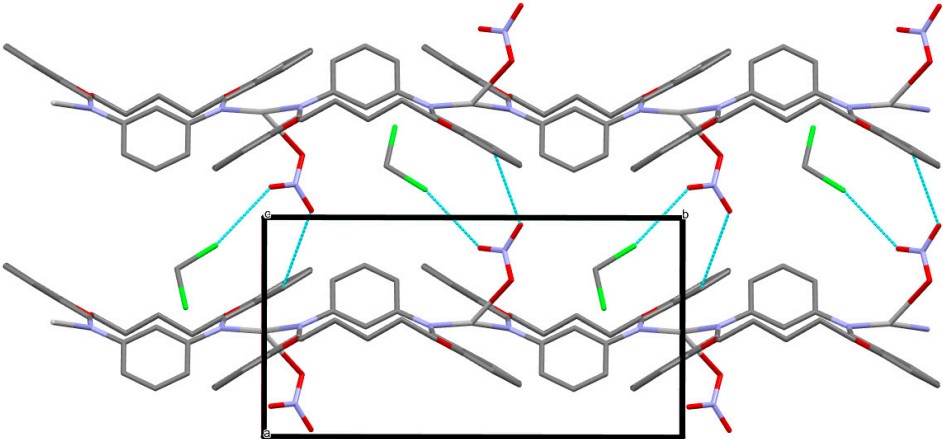

**Figure 7.** Representation of crystal packing in BUPREL. C–H···O and Cl···O interactions between anions, solvent and ligand molecules are represented by blue dashed lines. Hydrogen atoms are omitted for clarity. The view is down the *c* axis.

In the first step, formation of the polymeric chain through Ag-N bond formation occurs; in the second step, anions interact with adjacent polymeric chains (either as coordinated species or by intermolecular interactions), forming a 2D ribbon motif; in the final step, solvent molecules occupy residual voids (Scheme 2).

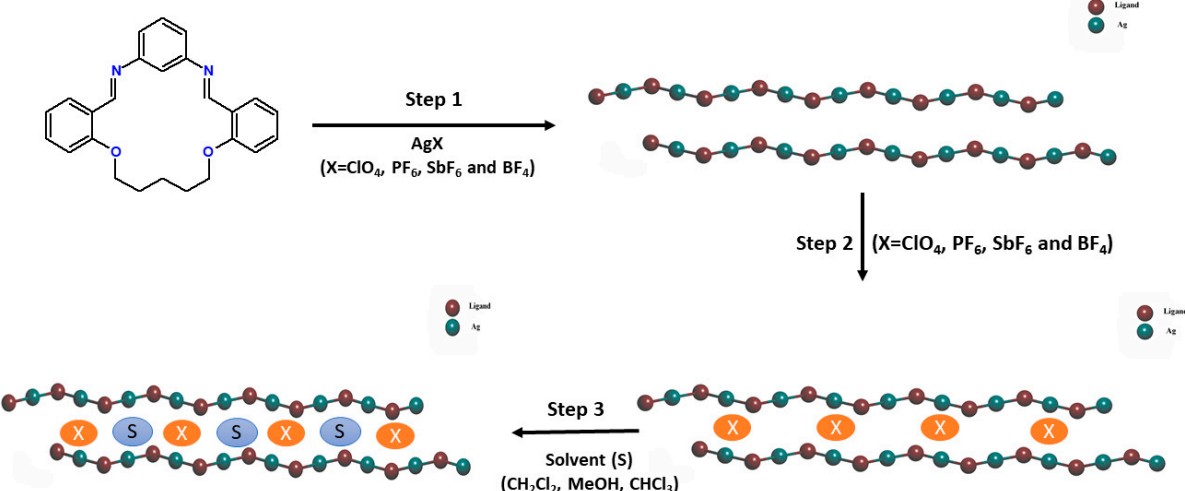

**Scheme 2.** A plausible 3-step mechanism for the formation of compounds.

Anion impact on supramolecular assembly is most obvious and pronounced in the AgLSbF$_6$ compound. In this compound, the ligand molecule changes its conformation from non-planar to planar molecule due to the presence of a bulky octahedral anion (see

Table S5 for details regarding ligand conformation). The consequence of this conformational change is the formation of aromatic interactions (offset $\pi\cdots\pi$ interactions) between adjacent polymeric chains. Unlike in previous structures, in which anions and solvent molecules serve as a bridge between chains, here these interactions can be considered as the primary mode of interchain connection, and anions act as an auxiliary unit in the formation of the final 3D structural motif.

*3.4. Thermal Analysis (TG/DSC)*

The samples were treated in a reactive oxygen atmosphere up to 550 °C. The measurements were performed on samples that were filtered from a U-shaped tube, and since the crystals of the samples disintegrate under normal conditions, we assume that the solvent molecules from the compounds have evaporated. TG and DSC curves are represented in Figure S7. All attempts to measure the TG of AgLClO$_4$, even in an inert atmosphere, failed due to the explosive nature of the compound. The compound AgLBF$_4$ decomposes in three distinct steps. The first step occurs in the temperature interval from 120 to 208 °C and is accompanied by a mass loss of 11%. The corresponding mass loss and thermal event can be attributed to the decomposition of the BF$_4$ anion (calc. 14%). The following decomposition steps occur in temperature intervals from 240 to 500 °C and are accompanied by a total mass loss of 61%. These two events can be assigned to the decomposition of ligand molecules. The calculated percentage of Ag in the complex is 19%. The final residue (23%) is most probably composed of a mixture of elemental Ag and Ag$_2$O, and due to the presence of the BF$_4$ anion, the presence of B$_2$O$_3$ cannot be excluded. This could also be an explanation for discrepancies in calculated and observed weight losses. AgLPF$_6$ decomposes in two steps. The first step occurs in the interval from 240 to 500 °C and is accompanied by a mass loss of 23%, which agrees with the fraction of PF$_6$ anions in the structure (22%). The second thermal event occurs in the interval from 320 to 550 °C and can be assigned to the decomposition of the ligand molecule. The AgSbF$_6$ compound is thermally stable up to 300 °C; after that, a gradual mass loss can be observed. This mass loss can be assigned to the decomposition of the ligand molecule. A very abrupt thermal event was observed at 475 °C, accompanied by a strong exothermic event on the DSC curve and a mass loss of 32%. This event can be attributed to the decomposition of the SbF6 anion (calc. 32%). Such an abrupt thermal event and subsequent drift on the balance could imply that the material is potentially explosive; therefore, special care should be taken when handling this sample. The final residue of 23% can be assigned to a mixture of silver and antimony oxides. From the obtained data, the following thermal stability series can be constructed: AgSbF$_6$ > AgLPF$_6$ > AgLBF$_4$, indicating the importance of anions in the overall stability of crystal structures. In addition, the temperature of the thermal decomposition of ligand molecules in AgSbF$_6$ is somewhat higher than in other compounds, which could be a consequence of ring flattening and the formation of aromatic interactions (vide supra).

**4. Conclusions**

Four novel silver(I) complexes of aza-oxa macrocyclic Schiff base L with four different silver salts were prepared. The coordination geometry in compounds is linear (AgLClO$_4$, AgLBF$_4$ and AgLSbF$_6$) and T-shaped (AgLPF$_6$). Such a coordination mode results in the formation of infinite 1D polymeric chains displaying zigzag topology. Compounds AgLClO$_4$ and AgLBF$_4$ were found to be isostructural. In AgLPF$_6$, there are three symmetrically unique Ag ions, two of which are linearly coordinated with ligand molecules and form a discrete polymer chain, whereas the third silver ion is coordinated with two ligand molecules and a methanol molecule in a T-shaped geometry. Due to the presence of a bulky octahedral SbF$_6$ anion in the AgLSbF$_6$ compound, the ligand molecule changed conformation from non-planar to planar (also known as conformational switching). The result of this conformational change is the formation of offset $\pi\cdots\pi$ interactions between adjacent ligand molecules. A detailed analysis of intermolecular interactions indicates that the type and form of these interactions mostly depend on the type of anion. A supramolec-

ular motif observed in all compounds (including previously published nitrate analog) can be described as 2D sheets of ligand–metal polymeric chains with anions and solvent molecules sandwiched between them. Compound $AgLPF_6$ is unique regarding the presence of methanol as a solvent molecule between adjacent chains. We have concluded by a simple calculation that due to the size of the solvent-accessible void, only methanol molecules can be included in the crystal structure. These observations also implicate a possible 3-step mechanism for the formation of structures: in the first step, formation of a polymeric chain occurs through Ag-N bond formation; in the second step, anions occupy spaces between adjacent chains and form interactions with adjacent chains; and in the final step, solvent molecules occupy residual voids. The final step could be discriminatory towards solvents by the size of a solvent molecule. The obtained compounds were characterized by IR spectroscopy, which showed maxima typical for this type of CPs and included anions. The TG analysis indicated considerable thermal stability of prepared compounds, with some compounds thermally stable over 300 °C. The present study has shown that anions can have a significant impact on the supramolecular assembly of CPs. It is also worth noting that the ligand molecule can adopt conformation when forced by the presence of a bulky anion. These conformational changes are subtle but have a significant impact on interchain interactions. Most importantly, this research—although strictly speaking, it belongs to structural chemistry—gives a very interesting insight into the possible mechanism of the formation of CPs and therefore represents an important contribution to this particular field of research. These compounds can also be considered potential APIs due to the presence of silver in the molecular structure. We have attempted to determine some biological properties, but unfortunately, the solubility of compounds is very poor in most solvents, and therefore we were not able to carry out these tests. Despite this, we will devote our future research in this area to the search for soluble silver CPs with these types of macrocyclic ligands and to the examination of biological properties.

**Supplementary Materials:** The following supporting information can be downloaded at: https://www.mdpi.com/article/10.3390/cryst13010050/s1, Figure S1. IR spectrum of $AgLBF_4$, Figure S2. IR spectrum of $AgLClO_4$, Figure S3. IR spectrum of $AgLPF_6$, Figure S4. IR spectrum of $AgLSbF_6$, Table S1. Selected interatomic bond distances (Å) and valence angles (°) for compound $AgLClO_4$, Table S2. Selected interatomic bond distances (Å) and valence angles (°) for compound $AgLPF_6$, Table S3. Selected interatomic bond distances (Å) and valence angles (°) for compound $AgLBF_4$, Table S4. Selected interatomic bond distances (Å) and valence angles (°) for compound $AgLSbF_6$, Table S5. Conformational parameters for ligand and silver complexes, Figure S5. Structure overlay of $AgLSbF_6$ (blue) and $AgLClO_4$ (red), Figure S6. *trans* and *cis* configured polymeric chains in $AgLSbF_6$ (a) and $AgLClO_4$ (b), Figure S7. TG and DSC curves of $AgLSbF_6$ (blue), $AgLBF_4$ (black) and $AgLPF_6$. CCDC 2223300–2223303 contains the supplementary crystallographic data for this paper. These data can be obtained free of charge via http://www.ccdc.cam.ac.uk/conts/retrieving.html, accessed on 19 December 2022, or from the Cambridge Crystallographic Data Centre, 12 Union Road, Cambridge CB2 1EZ, UK; fax: (+44) 1223-336-033; or e-mail: deposit@ccdc.cam.ac.uk.

**Author Contributions:** Conceptualization, funding acquisition, supervision, writing—original draft preparation, writing—review and editing, and investigation, T.B.; writing—original draft preparation, writing—review and editing, formal analysis, and investigation, F.P.; formal analysis, writing—original draft preparation, and investigation, L.Š.; formal analysis, writing—original draft preparation, and investigation, M.K. All authors have read and agreed to the published version of the manuscript.

**Funding:** This work was carried out with financial support from Josip Juraj Strossmayer University of Osijek (Project: UNIOS-ZUP 2018-112) and from the Slovenian Research Agency (ARRS) (Project P1-0175).

**Institutional Review Board Statement:** Not applicable.

**Informed Consent Statement:** Not applicable.

**Data Availability Statement:** The data used in this study is available upon request.

**Conflicts of Interest:** The authors declare no conflict of interest.

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
