# Peer review of "Anion Influence on Supramolecular Interactions in Exo-Coordinated Silver(I) Complexes with N2O2 Schiff Base Macrocycle"

_crystals, doi:10.3390/cryst13010050_

Round 1
Reviewer 1 Report
The manuscript “Anion influence on supramolecular interactions in exo-coordinated silver(I) complexes with N2O2 Schiff base macrocycle” by T. Balić et al. describes the synthesis and structural characterization of four Ag(I) complexes. The manuscript is comprehensive and as an overall well written, but some aspects need improvement.
Lines 42-43 – What do you mean by “the characteristic of all macrocyclic complexes is great thermodynamic and kinetic stability”?
Line 72 – A reference is required for “CSD code BUPREL”
Line 294 – BUPREL structure is discussed in section 3.3 and reference to hydrogen bridging is mentioned without a scheme of the structure. Such scheme would help comprehension.
Lines 388-389 – correct according to what is written in lines 320-322. As it is wrong!
Exo and endo as well as cis and trans must be written in italic.
A question remains:
Which are possible applications of this type of coordination polymers?
Author Response
Dear colleague,
Thank you very much for your comments. We have corrected all of comments and responses are attached.
Best wishes,
Tomislav Balic

Reviewer 2 Report
Dear Prof. Ms. Ploy Assavajamroon
Assistant Editor Crystals
Thank you very much for choosing me as a potential reviewer for the Manuscript Title “Anion influence on supramolecular interactions in exo-coordinated silver(I) complexes with N2O2 Schiff base macrocycle "
Comments
1. The paper is well-organized.
2. Please check the English of all manuscript.
3. In the Introduction part, if possible add paragraph about the Schiff base ligand (articles for example https://doi.org/10.1002/aoc.617 and https://doi.org/10.3390/inorganics10110177) and the rationale behind the choice of the metal ion (sliver(I)) used for the synthesis of the complexes should be discussed.
4. In section Material and method the authors mentioned that they have used reagents and solvents already purchased, but without offering any names, values of purity, and so on. Please insert details about them and the suppliers. Is not sufficient only to mention in the general mode.
5. Discussion of the spectroscopic data should be more concise and the obtained data should be compared with those for the structurally similar complexes.
6. Therefore represents an important contribution to this particular field of research……..??? what about the applications?
According to the above comments, this paper may be accepted for publishing in Crystals after revision.
Author Response
Dear colleague,
Thank you for your comments. We have corrected all comments and responses are in the attachment.
Best wishes
